# Obesity, Physical Performance, Balance Confidence, and Falls in Community-Dwelling Older Adults: Results from the Korean Frailty and Aging Cohort Study

**DOI:** 10.3390/nu16050614

**Published:** 2024-02-23

**Authors:** Ga Yang Shim, Myung Chul Yoo, Yunsoo Soh, Jinmann Chon, Chang Won Won

**Affiliations:** 1Department of Physical and Rehabilitation Medicine, Kyung Hee University College of Medicine, Kyung Hee University Hospital, Seoul 02447, Republic of Korea; wholhear@gmail.com (G.Y.S.); famousir@naver.com (M.C.Y.); soyuns@gmail.com (Y.S.); 2Department of Family Medicine, Kyung Hee University College of Medicine, Kyung Hee University Hospital, Seoul 02447, Republic of Korea

**Keywords:** obesity, central obesity, falls, fall-related fractures, balance confidence

## Abstract

Obesity affects physical functions in numerous ways. We aimed to evaluate the association between obesity and falls, physical performance, and balance confidence in community-dwelling older adults. Using first-year baseline data from the Korean Frailty and Aging Cohort Study, 979 older adults were included. General obesity was defined based on the body mass index and body fat percentage, whereas central obesity was classified based on the waist circumference and waist-to-height ratio. Data regarding fall history and balance confidence were acquired using self-questionnaires, and a timed up-and-go test was performed to measure balance-related physical performance. Overall, 17.3% of participants experienced falls in the previous year. Central obesity, as determined by waist circumference (odds ratio, 1.461; 95% confidence interval, 1.024–2.086; *p*-value, 0.037) and by waist-to-height ratio (odds ratio, 1.808; 95% confidence interval, 1.015–3.221; *p*-value, 0.044) was significantly associated with falls. Interestingly, general obesity, measured by body fat percentage, was protective against fall-related fractures (odds ratio, 0.211; 95% confidence interval, 0.072–0.615; *p*-value, 0.004). Participants with central obesity had poorer physical performances in the timed up-and-go test (odds ratio, 2.162; 95% confidence interval, 1.203–3.889; *p*-value, 0.010) and lower balance confidence according to the Activities-specific Balance Confidence scale (odds ratio, 1.681; 95% confidence interval, 1.153–2.341; *p*-value 0.007). In conclusion, assessment of central obesity, particularly waist circumference, should be considered as a screening strategy for falls, and older adults with a high waist circumference should receive advice on fall prevention.

## 1. Introduction

Falls and fall-related injuries are common among older adults and are leading causes of morbidity and disability [1,2,3]. The prevalence of falls was reported to be 13–38% among community-dwelling older Korean adults [4,5]. Considering that Korea has one of the fastest-aging populations among countries worldwide, falls and medical costs due to falls are expected to continue rising in the future [6]. According to the recent literature, the risk factors for falls are diverse and can be classified as intrinsic (e.g., age, female sex, cognitive abilities, fear of falling, and chronic diseases) and extrinsic factors (e.g., poor-fitting footwear, slippery floor, or loose rugs) [7]. Some of these are modifiable risk factors and should be targeted to prevent falls.

Obesity is considered an individual risk factors for falls; however, conflicting evidence exists regarding its effect on falls. Some studies have reported that obesity is associated with an increased risk of falls in older people [8,9,10]. However, a study by Cho et al. showed that general obesity, as determined by body mass index (BMI), has no significant association with fall risk, whereas central obesity, based on waist circumference (WC), is a risk factor for falls among older adults [11]. The reason for these conflicting results may stem from different definitions of obesity.

Obesity is largely divided into general and central obesity. The former is commonly defined using BMI, with a cutoff point of 25 kg/m^2^ in Korea [12]. However, BMI might be a poor indicator of adiposity in older adults because it does not distinguish between adiposity and muscle [13,14]. Alternatively, the percentage of body fat (PBF), measured using dual-energy X-ray absorptiometry (DXA), has emerged as a more reliable indicator of general obesity [15]. Meanwhile, central obesity refers to the excessive deposition of intra-abdominal fat, and waist circumference (WC) measurements are considered one of the best and easiest methods for assessing the central distribution of body fat. However, such measurements are inaccurate in diagnosing obesity in people with high masses. For this purpose, the waist-to-height ratio (WHtR) has been proposed [16], which was a more favorable indicator of metabolic and cardiovascular diseases than WC [17,18]. 

The aim of this study was to investigate the association between general or central obesity and falls and fall-related fractures and to determine the effects of general and central obesity on balance confidence and physical performance.

## 2. Materials and Methods

### 2.1. Study Population

In total, 1559 community-dwelling older adults aged ≥ 70 were recruited. The data used in this study were obtained from the Korean Frailty and Aging Cohort Study (KFACS). The KFACS is a multi-institute, cross-sectional study that began in 2016 to identify and prevent the factors contributing to negative outcomes related to aging in the community-dwelling older population.

Among the 1559 participants, those with body composition data measured using whole-body DXA were included. Participants with a cerebrovascular accident history, paraplegia or tetraplegia, dementia, a Korean Mini-Mental State Examination score < 23, or blindness in one or both eyes were excluded. Finally, 979 participants (483 men and 496 women) were included (Figure 1). 

This research was supported by a grant from the Korea Health Technology R&D Project through the Korean Health Industry Development Institute, funded by the Ministry of Health and Welfare, Republic of Korea (grant no. HI15C3153). This study was approved by the institutional review board of Kyung Hee University Hospital (approval no., KHUHMDIRB 2015-12-103; approval date, 19 May 2016). All participants signed an informed consent form before participation.

### 2.2. General Obesity and Central Obesity

General obesity was defined using two indices, BMI and PBF. Weight was assessed using a portable digital scale with a precision of 0.1 kg while standing height was measured with a tape to the nearest 0.1 cm. BMI is calculated as weight divided by the square of height. In Korea, individuals with a BMI equal to or greater than 25 kg/m^2^ are classified as obese [12]. PBF was calculated as the ratio of total fat mass to weight multiplied by 100. Body fat mass was measured using DXA systems from Lunar (GE Healthcare, Madison, WI, USA) and Hologic (Hologic Inc., Bedford, MA, USA). The participants were asked to remove all metal accessories and lie supine on the scanner table with limbs placed parallel to their bodies, following the manufacturer’s protocol [19]. General obesity was defined as a PBF of 25 or higher for men and 35 or more for women [20].

Central obesity was determined based on WC and WHtR. The WC was measured at the midpoint between the lower end of the last rib and the upper ridge of the iliac crest using an inelastic measuring tape with a precision of 0.1 cm [21]. The WC cutoffs for Korean adults are ≥90 cm for men and ≥85 cm for women [22]. The WHtR cutoff value for defining central obesity is 0.50 for both sexes [23]. Central obesity was categorized as follows: (1) no central obesity (WC < 90 cm for men and <85 cm for women) or (2) central obesity (WC ≥ 90 cm for men and ≥85 cm for women). Additionally, it was classified based on WHtR as follows: (1) no central obesity (WHtR < 0.50) or (2) central obesity (WHtR ≥ 0.50 for both sexes).

### 2.3. History of Falls, Fall-Related Fractures, Physical Performance, and Balance Confidence

A fall was defined as an event involving the unintentional contact of any part of the body with the ground or a lower level [24]. Fall and fall-related fractures were investigated through a self-questionnaire as follows, which included the following questions: “How many times have you fallen in the last 12 months?” and “If you fell, did you have any fractures?”

The timed up-and-go (TUG) test is a commonly used screening tool for assessing fall risks. The TUG test has been shown to have good test–retest reliability for community-dwelling older adults [25]. Participants were asked to sit on a chair, stand up, walk 3 m at a comfortable pace, navigate around an obstacle, and return to sit on the chair. The examiner measured the time taken to complete the task [26]. Faster completion times indicate better functional performance. A time of ≥13.5 s was used as a reference point to identify those at a high risk of falls among community-dwelling older adults [27]. With this reference point, the participants were divided into poor and good physical performance groups. 

Balance confidence was assessed using the Activities-specific Balance Confidence (ABC) scale. The ABC scale consisted of 16 items, and participants were asked to score how confident they were in performing each task [28]. Each task score ranges from 0% to 100%, and the total score was obtained by averaging each item score. A higher percentage indicates that the participant has a greater degree of balance confidence. A previous study suggested that an ABC score of <67% reliably predicts future falls [29]. Therefore, we divided the participants into ‘low’ and ‘high’ performance groups using this cutoff. The ABC scale was shown to be a reliable and valid method for measuring balance confidence in older Korean adults [30].

### 2.4. Other Parameters

Data on sociodemographic and health-related variables were acquired using self-administered questionnaires. Sociodemographic variables included age, sex, education level (no formal education, elementary school, middle/high school, and college/university), and marital status (married and single/never married/divorced/widowed). Health-related factors included current smoking (yes or no) and alcohol consumption frequency exceeding five times per week (yes or no). A medical history of hypertension, dyslipidemia, diabetes, and osteoarthritis was also obtained. Blood samples were obtained at approximately 8 AM after an overnight fast of at least 8 h and were analyzed as previously described [31].

### 2.5. Statistical Analysis

Continuous variables were described using means and standard deviations, and categorical variables were described using frequencies and percentages. We investigated the normality of continuous variables using the Kolmogorov–Smirnov test. Baseline characteristics were reported and compared in accordance with fall history using the chi-square test for categorical variables and independent t-tests or the Mann–Whitney U test for continuous variables. Logistic regression analysis was conducted to examine the associations between obesity and a history of falls, physical performance, and balance confidence. Covariates related to falls in this study (age, sex, marital status, education level, and osteoarthritis) were adjusted in a logistic regression model. Results were represented as the odds ratio (OR) and 95% confidence interval (95% CI). Statistical analyses were carried out using the Statistical Package for the Social Sciences (SPSS) version 23.0 for Windows (IBM Corp., Armonk, NY, USA). A *p*-value of <0.05 was considered indicative of statistical significance.

## 3. Results

### 3.1. Characteristics of Participants

In total, 979 participants were enrolled in this study. The mean age was 75.9 ± 3.9 years, and 50.7% were women. The prevalence of general obesity, defined based on BMI and PBF, was 40.7% and 67.0%, respectively, whereas the prevalence of central obesity, defined based on WC and WHtR, was 53.4% and 84.6%, respectively. The proportion of participants who experienced one or more falls in the preceding year was 17.3% (169/979). Compared with the group not experiencing falls, the group experiencing falls had a higher proportion of females, non-married individuals, and individuals with a low education level. Regarding obesity indicators, PBF, WC, and WHtR were significantly higher in the faller group than participants who did not experience falls (Table 1).

### 3.2. Association of Falls and Fall-Related Fractures with General Obesity and Central Obesity

Table 2 shows the logistic regression analysis results of obesity indicators according to falls and fall-related fractures. Although no association was observed between general obesity and falls after multivariate adjustment, general obesity, defined based on PBF, was associated with a reduced OR for fall-related fractures (OR, 0.211; 95% CI, 0.072–0.615; *p*-value, 0.004). Central obesity, based on WC and WHtR, was significantly associated with falls in the multivariate-adjusted model (OR, 1.461; 95% CI, 1.024–2.086, *p*-value, 0.037 for WC; OR, 1.808; 95% CI 1.015–3.221; *p*-value 0.044 for WHtR).

### 3.3. Association of Balance-Related Physical Performance with General Obesity and Central Obesity

Table 3 presents the distribution and ORs concerning physical performance in relation to indicators of obesity. The poor physical performance group had a higher proportion of individuals with general obesity, based on PBF, and central obesity, defined based on WC and WHtR. In the multivariate logistic regression analysis, participants with central obesity, defined based on WC, demonstrated an OR of 2.162 (95% CI, 1.203–3.889; *p*-value, 0.010) for poor physical performance.

### 3.4. Association of Balance Confidence with General Obesity and Central Obesity

Table 4 shows the distribution and ORs of balance confidence according to obesity indicators. Low balance confidence was more frequent in individuals with general obesity, based on PBF, and central obesity, defined based on WC and WHtR. In the multivariate logistic regression analysis, participants with central obesity, defined based on WC, had an OR of 1.681 (95% CI, 1.153–2.341; *p*-value, 0.007) for low balance confidence.

## 4. Discussion

Falls represent a significant challenge to healthy aging, and as life expectancy increases, the prevalence of falls is expected to increase. This study examined the associations of general obesity (based on BMI and PBF) and central obesity (based on WC and WHtR) with falls, physical performance, and balance confidence in community-dwelling older adults. The associations were only observed for central obesity. This suggests that an assessment of central obesity, notably WC, should be considered as a screening strategy for falls and that older adults with central obesity should receive advice on fall prevention. 

Several studies have investigated the association between general obesity and falls. However, for general obesity, defined based on BMI, inconsistent results have been obtained. Zhao et al. reported that higher BMI increases the OR for falls [32], whereas other studies showed a U-shape association between BMI and falls [33,34]. Conversely, Zhang et al. suggested that obesity is a protective factor against falls and hip fractures in nursing home residents [35]. Moreover, Hermenegildo-Lopez et al. concluded that general obesity, as defined based on BMI, is not significantly associated with falls [36]. Consistently, general obesity, defined based on both BMI and PBF, was not associated with falls in our study. 

Although BMI is widely used to determine obesity, one limitation is that BMI does not reflect body composition [37]. With aging, body composition is known to be altered based on muscle loss and increased body fat [38]. For example, one individual could have an excessive amount of body fat, and another might have significantly more lean body mass; however, since BMI is weight divided by height squared, the two individuals could have the same BMI. Therefore, in the aforementioned studies, the high-BMI group could have included a mix of individuals with either high body fat or high lean body mass, which might have led to contradictory results. 

Meanwhile, general obesity is believed to exhibit a positive relationship with fall-related fractures. A Spanish study found that general obesity has a protective effect against falls requiring medical care or falls associated with fractures [36]. Our study also showed that general obesity determined based on PBF was found to have a protective effect against fall-related fractures. One possible explanation for this protective effect is that the fatty tissue found around the hips would have a cushioning effect during falls, reducing the risk of fracture. Additionally, higher body fat mass increases the bone load, which increases bone mineral density and reduces the risk of fractures. 

Central obesity is characterized by increased abdominal fat mass. In particular, as body fat is redistributed with age, intra-abdominal fat increases more than subcutaneous fat. Choi et al. showed that central obesity can predict falls among community-dwelling American adults ≥ 65 years of age [11]. Further, Masimo et al. found that central obesity is positively associated with single and recurrent falls in Brazilians 60 years of age and older [39]. Meanwhile, Kioh et al. showed that a higher waist-to-hip ratio, an indicator of central obesity, remains independently associated with an increased risk of falls compared to the risk in individuals with a lower waist-to-hip ratio [40]. Consistent with these findings, the present study showed that central obesity, defined based on both WC and WHtR, is associated with a higher risk of falls compared to non-central obesity. 

The consistent association between central obesity and falls might be explained by altered body geometry. As abdominal fat increases, the body’s center of mass shifts forward, and the anterior pelvic tilt increases, resulting in postural instability [41,42]. Central obesity also affects gait biomechanics: older adults with obesity had a shorter swing phase, longer stance phase, and shorter single support phase than the non-obesity group [43]. Such postural instability and changes in gait patterns increase the risk of falls, decreased physical performance, and poor balance confidence.

The prevalence of obesity depends on its definition in our study; the prevalence based on PBF was 67.0%, whereas the prevalence based on BMI was 40.7%. This difference suggests that BMI underestimates obesity in older adults because height and weight do not reflect changes in fat gain and muscle loss with age. Central obesity defined based on WHtR had a higher prevalence than central obesity based on WC, at 53.4% and 84.6%, respectively. WHtR tends to overestimate central obesity in older adults, which can be explained by over-adjustment of the ratio as height decreases with age as a result of decreased intervertebral disc height or compression fractures [44]. 

The TUG test, which involves sitting down on a chair, standing up, walking 3 m, and returning to avoid an obstacle, can be used to assess not only mobility but also dynamic balance. The TUG test is known as a physical performance assessment that can predict falls [45]. In this study, the central obesity defined based on WC showed that there were fewer participants who completed the TUG test quickly compared to the normal group. These results are consistent with a previous study showing that abdominal obesity is associated with reduced gait speed in adults [23]. The balance confidence scale, ABC, is a recognized fall risk screening tool [28]. In the present study, a significantly lower balance confidence was observed in the central obesity group defined using WC. Individuals with a higher WC experience difficulty in bending, kneeling, stooping, lifting, and carrying [24], which could influence balance confidence when performing functional tasks. A simple clinical measurement of WC might thus be useful in predicting poor physical performance and low balance confidence.

Physical activity and sedentary time are associated with the risk of falls. Prolonged sitting and low physical activity detrimentally impact muscle health in older adults, leading to diminished muscle strength and function, thereby heightening the risk of falls [46]. Interestingly, older adults with obesity tend to be less physically active and have shorter exposure times, which might reduce the risk of falls [47]. We did not consider the physical activity level or the amount of sedentary time, which could have influenced the results. Future studies should address these potential confounding factors.

Several limitations are noteworthy in this study. Firstly, the history of falls was self-reported, and it is susceptible to recall bias. Secondly, as this was a cross-sectional study, it cannot be used to infer a causal association between the variables and the occurrence of falls. Thirdly, as only adults aged 70 years or older were included, the results of this study cannot be generalized to the entire population. Last, our study did not consider nutritional status, which affects obesity.

## 5. Conclusions

In conclusion, older adults with central obesity showed a higher risk of experiencing falls within the past year. Likewise, those with a higher WC were more prone to having reduced balance confidence and impaired physical performance compared to those with a lower WC. Therefore, evaluating central obesity, particularly through WC measurements, should be considered as a screening strategy for falls, and older adults with a high WC should receive advice on fall prevention.

## Figures and Tables

**Figure 1 nutrients-16-00614-f001:**
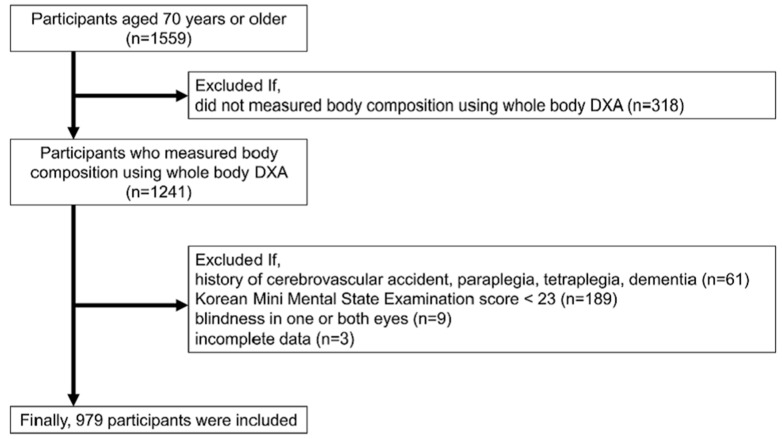
Flow chart of study participants.

**Table 1 nutrients-16-00614-t001:** Characteristics of study participants according to fall history.

	Total (*n* = 979)	Faller (*n* = 169)	Non-Faller (*n* = 810)	*p*-Value
Age (years)	75.9 ± 3.9	76.1 ± 3.6	75.8 ± 3.4	0.001 ^†^
Female	496 (50.7)	106 (62.7)	390 (48.1)	0.001
Marital status				<0.001
Married	665 (67.9)	93 (55.0)	572 (70.1)	
Single/never married/divorced/widowed	314 (36.7)	76 (43.0)	238 (29.4)	
Education level				0.004
No formal education	131 (13.4)	32 (18.9)	99 (12.2)	
Elementary school	265 (27.1)	48 (28.4)	217 (26.8)	
Middle/high school	384 (39.2)	66 (39.1)	318 (39.3)	
College/university	199 (20.3)	23 (13.6)	176 (21.7)	
Current smoker	49 (5.0)	7 (4.1)	42 (5.2)	0.574
Current alcohol	521 (53.2)	84 (49.7)	437 (54.0)	0.322
Height (cm)	158.5 ± 8.8	156.5 ± 8.4	158.9 ± 8.6	0.001
Weight (kg)	61.4 ± 9.5	60.5 ± 9.7	61.7 ± 9.4	0.154
Triglyceride (mg/dL)	119.6 ± 60.6	120.5 ± 59.8	119.5 ± 60.8	0.449 ^†^
HDL (mg/dL)	52.8 ± 14.1	54.2 ± 14.5	52.4 ± 14.0	0.124 ^†^
Fasting blood glucose (mg/dL)	104.5 ± 21.9	105.0 ± 21.0	104.4 ± 22.0	0.669 ^†^
Comorbidities				
Hypertension	555 (56.7)	95 (56.2)	460 (56.7)	0.904
Dyslipidemia	313 (32.0)	56 (33.1)	257 (31.7)	0.734
Diabetes	203 (20.7)	39 (23.1)	164 (20.2)	0.581
Osteoarthritis	223 (22.8)	51 (30.2)	172 (21.2)	0.029
General obesity by BMI	24.4 ± 3.0	24.6 ± 3.0	24.4 ± 3.0	0.340
Normal (BMI < 25)	581 (59.3)	97 (57.4)	484 (59.8)	0.562
Obese (BMI ≥ 25)	398 (40.7)	72 (42.6)	326 (40.2)	
General obesity by PBF	32.0 ± 7.9	33.7 ± 7.3	31.6 ± 8.0	0.002
Normal (men < 25, women < 35)	323 (33.0)	42 (24.8)	281 (34.7)	0.013
Obese (men ≥ 25, women ≥ 35)	656 (67.0)	127 (75.2)	529 (65.3)	
Central obesity by WC	87.9 ±8.4	88.7 ± 8.2	87.7 ± 8.4	0.154
No (men < 90, women < 85)	456 (46.6)	63 (37.3)	393 (48.5)	0.008
Yes (men ≥ 90, women ≥ 85)	523 (53.4)	106 (62.3)	417 (51.5)	
Central obesity by WHtR	0.56 ± 0.06	0.57 ± 0.05	0.55 ± 0.06	0.002
No (WHtR < 0.50)	151 (15.4)	15 (8.9)	136 (16.8)	0.010
Yes (WHtR ≥ 0.50)	828 (84.6)	154 (91.1)	674 (83.2)	

Values are mean standard deviation ± SD and number (percentage). Abbreviations: BMI, body mass index; HDL, high-density lipoprotein cholesterol; PBF, percentage of body fat; WC, waist circumference; WHtR, waist–height ratio. ^†^ *p*-value for Mann–Whitney test.

**Table 2 nutrients-16-00614-t002:** Association between falls and fall-related fractures and general and central obesity.

	Falls		Fall-Related Fractures	
	AOR (95% CI)	*p*-Value	AOR (95% CI)	*p*-Value
General obesity				
BMI (BMI ≥ 25)	1.029 (0.727–1.456)	0.872	0.974 (0.432–2.196)	0.950
PBF (men ≥ 25, women ≥ 35)	1.117 (0.735–1.885)	0.497	0.211 (0.072–0.615)	0.004
Central obesity				
WC (men ≥ 90, women ≥ 85)	1.461 (1.024–2.086)	0.037	1.215 (0.527–2.798)	0.647
WHtR (WHtR ≥ 0.50)	1.808 (1.015–3.221)	0.044	0.422 (0.159–1.120)	0.083

Abbreviations: AOR, adjusted odds ratio; CI, confidence interval; BMI, body mass index; PBF, percentage of body fat; WC, waist circumference; WHtR, waist–height ratio. Adjusted model: adjusted for age, sex, education level, marital status, and osteoarthritis.

**Table 3 nutrients-16-00614-t003:** Association between physical performance and general and central obesity.

	Physical Performance				
	Poor ^a^	Good ^b^	*p*-Value	AOR (95% CI)	*p*-Value
Obesity					
BMI (BMI ≥ 25)	27 (42.2)	371 (40.5)	0.791	1.059 (0.616–1.822)	0.835
PBF (men ≥ 25, women ≥ 35)	51 (79.7)	605 (66.1)	0.025	1.733 (0.819–3.669)	0.151
Central Obesity					
WC (men ≥ 90, women ≥ 85)	45 (70.3)	478 (52.2)	0.005	2.162 (1.203–3.889)	0.010
WHtR (WHtR ≥ 0.50)	57 (89.1)	771 (84.3)	0.306	1.328 (0.574–3.073)	0.508

Abbreviations: AOR, adjusted odds ratio; CI, confidence interval; BMI, body mass index; PBF, percentage of body fat; WC, waist circumference; WHtR, waist–height ratio. ^a^ Poor physical performance: TUG < 13.5 s. ^b^ Good physical performance: TUG ≥ 13.5 s. Adjusted model: adjusted for age, sex, education level, marital status, and osteoarthritis.

**Table 4 nutrients-16-00614-t004:** Association between balance confidence and general and central obesity.

	Balance Confidence				
	Low ^a^	High ^b^	*p*-Value	AOR (95% CI)	*p*-Value
Obesity					
BMI (BMI ≥ 25)	85 (46.7)	313 (39.3)	0.064	1.222 (0.855–1.751)	0.275
PBF (men ≥ 25, women ≥ 35)	149 (81.9)	507 (63.5)	<0.001	1.309 (0.778–2.203)	0.310
Central Obesity					
WC (men ≥ 90, women ≥ 85)	121 (66.5)	402 (50.4)	<0.001	1.681 (1.153–2.341)	0.007
WHtR (WHtR ≥ 0.50)	165 (90.7)	664 (83.2)	0.012	1.464 (0.812–2.646)	0.204

Abbreviations: AOR, adjusted odds ratio; CI, confidence interval; BMI, body mass index; PBF, percentage of body fat; WC, waist circumference; WHtR, waist–height ratio. ^a^ Low balance confidence: ABC score < 67%. ^b^ High balance confidence: ABC score ≥ 67%. Adjusted model: adjusted for age, sex, education level, marital status, and osteoarthritis.

## Data Availability

All cohort data supporting the findings of this study are accessible through the KFACS and are available to researchers upon reasonable request. Comprehensive information regarding published articles, news articles utilizing the KFACS database, data provision guidelines, and contact details can be found on the KFACS website.

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
