# Peer review of "Obesity, Physical Performance, Balance Confidence, and Falls in Community-Dwelling Older Adults: Results from the Korean Frailty and Aging Cohort Study"

_nutrients, 2024, doi:10.3390/nu16050614_

Round 1

Reviewer 1 Report

Comments and Suggestions for Authors

The research by Ga Yang Shim et al. aims to assess the association of obesity with falls, physical performance and balance confidence in community-dwelling older adults. I see this as interesting research for practical applications to improve the health of the population. The following are the improvements I consider necessary to improve the quality of the study.

- The introduction is well written, however I consider that the references are not up to date, if you look at number one it is from 1988, 36 years have passed, I doubt that there is no more research. The same is true for the rest of the references in that section, they can be updated.

- The same applies to the section on the differences between the types of obesity, which is a very well researched topic, so there are new references, I refer to point 2.2.

The references could also be improved in the discussion. The authors should reconsider the inclusion of Effects of obesity on gait biomechanics.

- lines 235 to 239 should be further elaborated on, including current references

- The conclusions need to be developed much further and a new section on practical applications needs to be included.

In general, I believe that the data provided are not up to date, there is a lack of research lines to contribute and to improve the sections that have been mentioned.

Author Response

Please see the attatchment

We hope the revised manuscript will better meet the requirements of the Nutrients. We thank you once again for the constructive review comments.

Reviewer 2 Report

Comments and Suggestions for Authors

This is an interesting manuscript using a large sample of Korean elderly participants to assess mainly the association between obesity (general and central) and falls. The content is original, and the discussion is well-argued. However, there are some weaknesses in the methodology.

Major issues:

My suggestions for improving the manuscript concern mainly three methodological aspects: 1- since anthropometric measurements constitute a major part of the study, it is necessary to report at least the reference manuals of anthropometric methodology and the measuring instruments used. 2- In the statistical analysis, you do not mention normality tests. Integrate this part. 3- The comparison in Table 1 considers the sample with combined sexes. This is not correct. As an alternative to the Student's t-test, I suggest you apply a two-way ANOVA so that you can also consider sex.

Minor issues: 

- line 49 and following: change “kg/m2” to “kg/m2”.

-line 67: specify how many participants had data with the DXA measurement.

-line 118: change “Height” to “Standing height”.

Comments on the Quality of English Language

None

Author Response

We hope the revised manuscript will better meet the requirements of the Nutrients. We thank you once again for the constructive review comments.

Round 2

Reviewer 2 Report

Comments and Suggestions for Authors

In general, I am satisfied with the changes made by the authors that helped improve the study even though the use of a mixed-sex sample is not entirely adequate.

Author Response

We appreciate the reviewer's valuable comments.